# REWEIGHTED PROXIMAL PRUNING FOR LARGE-SCALE LANGUAGE REPRESENTATION

## ABSTRACT

Recently, pre-trained language representation flourishes as the mainstay of the natural language understanding community, e.g., BERT. These pre-trained language representations can create state-of-the-art results on a wide range of downstream tasks. Along with continuous significant performance improvement, the size and complexity of these pre-trained neural models continue to increase rapidly. Is it possible to compress these large-scale language representation models? How will the pruned language representation affect the downstream multi-task transfer learning objectives? In this paper, we propose Reweighted Proximal Pruning (RPP), a new pruning method specifically designed for a large-scale language representation model. Through experiments on SQuAD and the GLUE benchmark suite, we show that proximal pruned BERT keeps high accuracy for both the pre-training task and the downstream multiple fine-tuning tasks at high prune ratio. RPP provides a new perspective to help us analyze what large-scale language representation might learn. Additionally, RPP makes it possible to deploy a large state-of-the-art language representation model such as BERT on a series of distinct devices (e.g., online servers, mobile phones, and edge devices).

## 1 INTRODUCTION

Pre-trained language representations such as GPT (Radford et al., 2018), BERT (Devlin et al., 2019) and XLNet (Yang et al., 2019), have shown substantial performance improvements using self-supervised training on large-scale corpora (Dai & Le, 2015; Peters et al., 2018; Radford et al., 2018; Liu et al., 2019a). More interestingly, the pre-trained BERT model can be fine-tuned with just one additional output layer to create state-of-the-art models for a wide range of tasks, such as question answering (Rajpurkar et al., 2016; 2018), and language inference (Bowman et al., 2015; Williams et al., 2017), without substantial task-specific architecture modifications. BERT is conceptually simple and empirically powerful (Devlin et al., 2019).

However, along with the significant performance enhancement, the parameter volume and complexity of these pre-trained language representations significantly increase. As a result, it becomes difficult to deploy these large-scale language representations into real-life computation constrained devices including mobile phones and edge devices. Throughout this paper, we attempt to answer the following questions.

**Question 1**: Is it possible to compress large-scale language representations such as BERT via weight pruning?

**Question 2**: How would the weight-pruned, pre-trained model affect the performance of the downstream multi-task transfer learning objectives?

The problem of weight pruning has been studied under many types of deep neural networks (DNNs) (Goodfellow et al., 2016), such as AlexNet (Krizhevsky et al., 2012), VGG (Simonyan & Zisserman, 2014), ResNet (He et al., 2016), and MobileNet (Howard et al., 2017). It is shown that weight pruning can result in a notable reduction in the model size. A suite of weight pruning techniques have been developed, such as non-structured weight pruning (Han et al., 2015), structured weight pruning (Wen et al., 2016), filter pruning (Li et al., 2016), channel pruning (He et al., 2017), ADMM-NN (Ren et al., 2019) and PCONV (Ma et al., 2019) to name a few. Different from pruning CNN-type models, BERT not only considers the metrics on the pre-training task, but also needs to make

allowance for the downstream multi-task transfer learning objectives. Thus, the desired weight pruning needs to preserve the capacity of transfer learning from a sparse pre-trained model to downstream fine-tuning tasks.

In this work, we investigate irregular weight pruning techniques on the BERT model, including the iterative pruning method (Han et al., 2015) and one-shot pruning method (Liu et al., 2019b). However, these methods fail to converge to a sparse pre-trained model without incurring significant accuracy drop, or in many cases do not converge at all (see supporting results in Appendix). Note that the aforementioned weight pruning techniques are built on different sparsity-promoting regularization schemes (Han et al., 2015; Wen et al., 2016), e.g., lasso regression ($\ell_1$ regularization) and ridge regression ($\ell_2$ regularization). We find that the failure of previous methods on weight pruning of BERT is possibly due to the inaccurate sparse pattern learnt from the simple $\ell_1$ or $\ell_2$ based sparsity-promoting regularizer. In fact, the difficulty of applying regularization to generate weight sparsity coincides with the observation in (Loshchilov & Hutter, 2018) on the imcompatibility of conventional weight decay ($\ell_2$ regularization) for training super-deep DNNs as BERT. It is pointed out that the main reason is that the direct optimization of a regularization penalty term causes divergence from the original loss function and has negative effect on the effectiveness of gradient-based update. To mitigate this limitation, (Loshchilov & Hutter, 2018) have modified the regularization in Adam by *decoupling weight decay regularization from the gradient-based update*, and have achieved state-of-the-art results on large-scale language pre-training and downstream multi-task transfer learning objectives (Devlin et al., 2019).

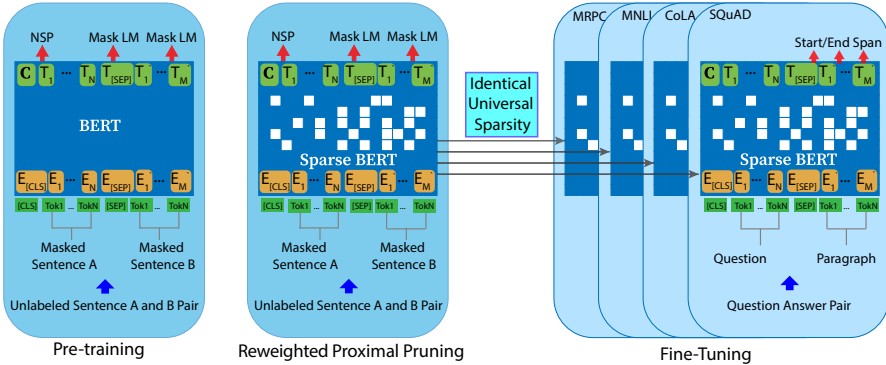

**Figure 1:** Overview of pruning BERT using Reweighted Proximal Pruning algorithm and then fine-tuning on a wide range of downstream transfer learning tasks. Through RPP, we find the identical universal sparsity $\mathcal{S}_{\tilde{\mathbf{w}}}$. The BERT model pruned with RPP could be fine-tuned over the downstream transfer learning tasks.

In this work, we aim at more accurate universal sparse pattern search (see Figure 1 for an overview of our approach) motivated by our experiments and the conclusion from Loshchilov & Hutter (2018). We propose Reweighted Proximal Pruning (RPP), which integrates reweighted $\ell_1$ minimization (Candes et al., 2008) with proximal algorithm (Parikh et al., 2014). RPP consists of two parts : the reweighted $\ell_1$ minimization and the proximal operator. Reweighted $\ell_1$ minimization serves as a better method of generating sparsity in DNN models matching the nature of weight pruning, compared with $\ell_1$ regularization. Thanks to the closed-form solution of proximal operation on a weighted $\ell_1$ norm, in RPP the sparsity pattern search can be decoupled from computing the gradient of the training loss. In this way the aforementioned pitfall in prior weight pruning technique on BERT can be avoided. We show that RPP achieves effective weight pruning on BERT for the first time to the best of our knowledge. Experimental results demonstrate that the proximal pruned BERT model keeps high accuracy on a wide range of downstream tasks, including SQuAD (Rajpurkar et al., 2016; 2018) and GLUE (Wang et al., 2018).

We summarize our contributions as follows.

- We develop the pruning algorithm Reweighted Proximal Pruning (RPP), which acheives the first effective weight pruning result on large pre-trained language representation model - BERT. RPP achieves $59.3\%$ weight sparsity without inducing the performance loss on both pre-training and fine-tuning tasks.

- We spotlight the relationship between the pruning ratio of the pre-trained DNN model and the performance on the downstream multi-task transfer learning objectives. We show that many downstream tasks except for SQuAD allows at least $80\%$ pruning ratio compared with $59.3\%$ under the more challenging task SQuAD.

- We observe that as the pruning ratio of the pre-trained language model increases, the performance on the downstream transfer learning tasks decreases. The descending range varies in different downstream transfer learning tasks. However, the proposed RPP approach is able to achieve a consistently high pruning ratio compared to iterative pruning based methods.

- We show that different from weight pruning in image classification tasks, RPP helps to find the structured sparsity pattern in transformer blocks used in BERT. Moreover, we peer into the effect of network pruning on the language representation embedded in BERT.

## 2    Related Work

**BERT and prior work on model compression**    BERT (Devlin et al., 2019) is a self-supervised approach for pre-training a deep transformer encoder (Vaswani et al., 2017), before fine-tuning it for particular downstream tasks. Pre-training of BERT optimizes two training objectives − masked language modeling (MLM) and next sentence prediction (NSP) − which require a large collection of unlabeled text. We use BooksCorpus (800M words) (Zhu et al., 2015) and the English instance of Wikipedia (2,500M words) as the pre-training corpus, the same as Devlin et al. (2019). For detailed information about the BERT model, readers can refer to the original paper (Devlin et al., 2019).

Michel et al. (2019) mask some heads in multi-head attention modules in BERT, and then evaluate the performance on the machine translation task. Similarly, Hao et al. (2019) eliminates certain heads in the multi-head attention module. First, the limited previous work do not consider the pre-training metrics and the other downstream multi-mask transfer learning objectives. They only considered the specific machine translation task (out of over 10 transfer tasks), which is only a specific fine-tuning and is limited for the universal pre-trained language representation (BERT). Second, the multi-head attention module uses a weight sharing mechanism (Vaswani et al., 2017). So masking some heads does not reduce the weight volume. Finally, multi-head attention allows the model to jointly attend to information from different representation subspaces at different positions, while single attention head inhibits this effect (Vaswani et al., 2017). As a result, masking some heads in multi-head attention harms the weight sharing mechanism, without weight volume reduction. In summary, the limited previous work in this area are not effective weight pruning method on BERT. Shen et al. (2019) reports the quantization result of BERT model, which is orthogonal to our work and can be combined for further compression/acceleration.

**Reweighted $\ell_1$ and proximal algorithm**    Candes et al. (2008) present reweighted $\ell_1$ algorithm and demonstrate the remarkable performance and broad applicability in the areas of statistical estimation, error correction and image processing. Proximal algorithms can be viewed as an analogous tool for non-smooth, constrained, large-scale, or distributed versions of these problems (Parikh et al., 2014). To the best of our knowledge, ours is the first work that applies reweighted $\ell_1$ minimization to network compression, particularly for BERT pruning. ,

## 3    Reweighted Proximal Pruning for large-scale language representation during pre-training

Pruning for pre-trained language representations should not only consider the performance of pre-training objectives, but also make allowance for the downstream fine-tuning transfer learning tasks. Let $f_i$ denote the loss function of network for downstream task $\mathcal{T}_i \sim p(\mathcal{T})$, where $p(\mathcal{T})$ denotes the distribution of tasks. Let $\mathbf{w}$ denote the parameters of the pre-trained model (pre-training in BERT), and $\mathbf{z}_i$ denote the $i$-th task-specified model parameters (fine-tuning in BERT). The downstream tasks have separate fine-tuned models, even though they are initialized with the same pre-trained parameters (Devlin et al., 2019). Starting from the pre-trained parameters $\mathbf{w}$, the parameters $\mathbf{z}_i(\mathbf{w})$ are obtained through fine-tuning

$$\underset{\mathbf{w} \in \mathbb{R}^d}{\text{minimize}} \; f_i(\mathbf{w}) \tag{1}$$

### 3.1 PRUNING FORMULATION IN TRANSFER LEARNING

Following the conventional weight pruning formulation, we first consider the problem of weight pruning during pre-training:

$$\underset{\mathbf{w} \in \mathbb{R}^d}{\text{minimize}} \; f_0(\mathbf{w}) + \gamma \|\mathbf{w}\|_p \tag{2}$$

where $f_0$ is the loss function of pruning, $p \in \{0, 1\}$ denotes the type of regularization norm, and $\gamma$ is a regularization term. We note that the sparsity-promoting regularizer in the objective could also be replaced with a hard $\ell_p$ constraint, $|\mathbf{w}\|_p \leq \tau$ for some $\tau$.

Let $\hat{\mathbf{w}}$ denote the solution to problem (2), and the corresponding sparse pattern $\mathcal{S}_{\hat{\mathbf{w}}}$ is given by

$$\mathcal{S}_{\hat{\mathbf{w}}} = \{i | \hat{w}_i = 0, \; \forall i \in [d]\} \tag{3}$$

For a specific transfer task $i$, we allow an additional retraining/fine-tuning step to train/fine-tune weights starting from the pre-training results $\hat{\mathbf{w}}$ and subject to the determined, fixed sparse pattern $\mathcal{S}_{\hat{\mathbf{w}}}$, denoted as $\mathbf{z}_i(\hat{\mathbf{w}}; \mathcal{S}_{\hat{\mathbf{w}}})$. That is, we solve the modified problem equation 1

$$\underset{\mathbf{z}_i}{\text{minimize}} \; f_i\big(\mathbf{z}_i(\hat{\mathbf{w}}; \mathcal{S}_{\hat{\mathbf{w}}})\big) \tag{4}$$

Here, different from (1), the task-specific fine tuning weights variable $\mathbf{z}_i(\hat{\mathbf{w}}; \mathcal{S}_{\hat{\mathbf{w}}})$ is now defined over $\mathcal{S}_{\hat{\mathbf{w}}}$.

Our goal is to seek a sparse (weight pruned) model during pre-training, with weight collection $\hat{\mathbf{w}}$ and sparsity $\mathcal{S}_{\hat{\mathbf{w}}}$, which can perform as well as the original pre-trained model over multiple new tasks (indexed by $i$). These fine-tuned models $\mathbf{z}_i(\hat{\mathbf{w}}; \mathcal{S}_{\hat{\mathbf{w}}})$ (for different $i$) share the *identical universal sparsity* $\mathcal{S}_{\hat{\mathbf{w}}}$.

### 3.2 REWEIGHTED PROXIMAL PRUNING

In order to enhance the performance of pruning pre-trained language representation over multi-task downstream transfer learning objectives, we propose Reweighted Proximal Pruning (RPP). RPP consists of two parts: the reweighted $\ell_1$ minimization and the proximal operator. Reweighted $\ell_1$ minimization serves as a better method of generating sparsity in DNN models matching the natural objective of weight pruning, compared with $\ell_1$ regularization. The proximal algorithm then separates the computation of gradient with the proximal operation over a weighted $\ell_1$ norm, without directly optimizing the entire sparsity-penalized loss, which requires gradient backpropagation of the involved loss. This is necessary in the weight pruning of super-deep language representation models.

#### 3.2.1 REWEIGHTED $\ell_1$ MINIMIZATION

In the previous pruning methods (Han et al., 2015; Wen et al., 2016), $\ell_1$ regularization is used to generate sparsity. However, consider that two weights $w_i, w_j$ ($w_i < w_j$) in the DNN model are penalized through $\ell_1$ regularization. The larger weight $w_j$ is penalized more heavily than smaller weight $w_i$ in $\ell_1$ regularization, which violates the original intention of weight pruning, "removing the unimportant connections" (parameters close to zero) (Han et al., 2015). To address this imbalance, we introduce reweighted $\ell_1$ minimization (Candes et al., 2008) to the DNN pruning domain. Our introduced reweighted $\ell_1$ minimization operates in a systematic and iterative manner (detailed process shown in Algorithm 1), and the first iteration of reweighted $\ell_1$ minimization is $\ell_1$ regularization. This designed mechanism helps us to observe the performance difference between $\ell_1$ and reweighted $\ell_1$ minimization. Meanwhile, this mechanism ensures the advancement of reweighted $\ell_1$ minimization over $\ell_1$ regularization, as the latter is the single, first step of the former.

Consider the regularized weight pruning problem (reweighted $\ell_1$ minimization):

$$\underset{\mathbf{w}}{\text{minimize}} \quad f_0(\mathbf{w}) + \gamma \sum_i \alpha_i |w_i| \tag{5}$$

where $\alpha_i$ ($\alpha_i > 0$) factor is a positive value. It is utilized for balancing the penalty, and is different from weight $w_i$ in DNN model. $\alpha_i$ factors will be updated in the iterative reweighted $\ell_1$ minimization procedure (Step 2 in Algorithm 1) in a systematic way (Candes et al., 2008). If we set $T = 1$ for reweighted $\ell_1$, then it reduces to $\ell_1$ sparse training.

---

**Algorithm 1** RPP procedure for reweighted $\ell_1$ minimization

---

1: Input: Initial pre-trained model $\mathbf{w}^0$, initial reweighted $\ell_1$ minimization ratio $\gamma$, initial positive value $\alpha^0 = 1$
2: **for** $t = 1, 2, \ldots, T$ **do**
3:     $\mathbf{w} = \mathbf{w}^{(t-1)}$, $\alpha = \alpha^{(t-1)}$
4:     **Step 1**: Solve problem (5) to obtain a solution $\mathbf{w}^t$ via iterative proximal algorithm (6)
5:     **Step 2**: Update reweighted factors $\alpha_i^t = \frac{1}{|w_i^t|^{(t)} + \epsilon}$ (the inside $w_i^t$ denotes the weight $w_i$ in iteration $t$, and the outside $(t)$ denotes the exponent), $\epsilon$ is a small constant, e.g., $\epsilon = 0.001$
6: **end for**

---

### 3.2.2 PROXIMAL METHOD

In the previous pruning methods (Han et al., 2015; Wen et al., 2016), $\ell_1$ regularization loss is directly optimized through the back-propagation based gradient update of DNN models, and the hard-threshold is adopted to execute the pruning action at the step of pruning (all weights below the hard-threshold become zero). In our approach, we derive an effective solution to problem (5) for given $\{\alpha_i\}$, namely, in Step 1 of Algorithm 2, in which back-propagation based gradient update is only applied on $f_0(\mathbf{w})$ but not $\gamma \sum_i \alpha_i |w_i|$.

We adopt the proximal algorithm (Parikh et al., 2014) to satisfy this requirement through decoupling methodology. In this way, the sparsity pattern search can be decoupled from back-propagation based gradient update of the training loss. The proximal algorithm is shown in (Parikh et al., 2014) to be highly effective (compared with the original solution) on a wide set of non-convex optimization problems. Additionally, our presented reweighted $\ell_1$ minimization (5) has analytical solution through the proximal operator.

To solve problem (5) for a given $\alpha$, the proximal algorithm operates in an iterative manner:

$$\mathbf{w}_k = \text{prox}_{\lambda_k, rw-\ell_1} \left( \mathbf{w}_{k-1} - \lambda_k \nabla_{\mathbf{w}} f_0 \left( \mathbf{w}_{k-1} \right) \right) \tag{6}$$

where the subscript $k$ denotes the time step of the training process inside each iteration of RPP, $\lambda_k$ $(\lambda_k > 0)$ is the learning rate, and we set the initial $\mathbf{w}$ to be $\mathbf{w}^{(t-1)}$ from the last iteration of reweighted $\ell_1$. The proximal operator $\text{prox}_{\lambda_k, rw-\ell_1}(\mathbf{a})$ is the solution to the problem

$$\underset{\mathbf{w}}{\text{minimize}} \, \gamma \sum_i \alpha_i \left| w_i \right| + \frac{1}{2\lambda_k} \| \mathbf{w} - \mathbf{a} \|_2^2 \tag{7}$$

where $\mathbf{a} = \mathbf{w}_{k-1} - \lambda_k \nabla_{\mathbf{w}} f \left( \mathbf{w}_{k-1} \right)$. The above problem has the following analytical solution (Liu et al., 2014)

$$w_{i,k} = \begin{cases} \left( 1 - \frac{\gamma \lambda_k \alpha_i}{|a_i|} \right) a_i & |a_i| > \lambda_k \gamma \alpha_i \\ 0 & |a_i| \leq \lambda_k \gamma \alpha_i. \end{cases} \tag{8}$$

We remark that the updating rule (6) can be interpreted as the proximal step (8) over the gradient descent step $\mathbf{w}_{k-1} - \lambda_k \nabla_{\mathbf{w}} f \left( \mathbf{w}_{k-1} \right)$. Such a descent can also be obtained through optimizers such as AdamW. We use the AdamW (Loshchilov & Hutter, 2018) as our optimizer, the same with (Devlin et al., 2019). The concrete process of AdamW with proximal operator is shown in Algorithm 3 of Appendix C.

*Why chooses AdamW rather than Adam?* Loshchilov & Hutter (2018) proposes AdamW to improve the generalization ability of Adam (Kingma & Ba, 2014). Loshchilov & Hutter (2018) shows that the conventional weight decay is inherently not effective in Adam and has negative effect on the effectiveness of gradient-based update, which is the reason of the difficulty to apply adaptive gradient algorithms to super-deep DNN training for NLU applications (like BERT). Loshchilov & Hutter (2018) mitigates this limitation and improves regularization of Adam, by decoupling weight decay regularization from the gradient-based update (Loshchilov & Hutter, 2018). AdamW is widely adopted in pre-training large language representations, e.g., BERT (Devlin et al., 2019), GPT (Radford et al., 2018) and XLNet (Yang et al., 2019). Our proposed RPP also benefits from the decoupling design ideology. The difference is that RPP is for the generation of sparsity, instead of avoiding over-fitting, like decoupled weight decay in AdamW.

**Our new and working baseline: New Iterative Pruning (NIP).** To get the *identical universal sparsity* $\mathcal{S}_{\mathbf{w}}$, we tried a series of pruning techniques, including the iterative pruning method (Han et al., 2015) and one-shot pruning method (Liu et al., 2019b). But these methods do not converge to a viable solution. The possible reason for non-convergence of the iterative pruning method is that directly optimizing $\ell_p$ ($p \in \{1, 2\}$) sparsity-promoting regularization makes the gradient computation involved and thus harms the loss convergence (We provide the loss curve and analysis in Appendix D). To circumvent the convergence issue of conventional iterative pruning methods, we propose a new iterative pruning (NIP) method. Different from iterative pruning (Han et al., 2015), NIP reflects the naturally progressive pruning performance without any externally introduced penalty. We hope that other pruning methods should not perform worse than NIP, otherwise, the effect of optimizing the newly introduced sparsity-promoting regularization is negative. We will show that NIP is able to successfully prune BERT to certain pruning ratios. We refer readers to Appendix A for the full detail about NIP, our proposed baseline algorithm.

## 4 EXPERIMENTS

In this section, we describe the experiments on pruning pre-trained BERT and demonstrate the performance on 10 downstream transfer learning tasks.

### 4.1 EXPERIMENT SETUP

We use the official BERT model from Google as the startpoint. Following the notation from Devlin et al. (2019), we denote the number of layers (i.e., transformer blocks) as $L$, the hidden size as $H$, and the number of self-attention heads as $A$. We prune two kinds of BERT model: $\text{BERT}_{\text{BASE}}$ ($L = 12, H = 768, A = 12$, total parameters $= 110\text{M}$) and $\text{BERT}_{\text{LARGE}}$ ($L = 24, H = 1024, A = 16$, total parameters $= 340\text{M}$). As the parameters of these transformer blocks take up more than 97% weights of the entire BERT, the weights of these transformer blocks are our pruning target.

**Data:** In pre-training, we use the same pre-training corpora as Devlin et al. (2019): BookCorpus (800M words) (Zhu et al., 2015) and English Wikipedia ($2,500\text{M}$ words). Based on the same corpora, we use the same preprocessing script[1] to create the pre-training data. In fine-tuning, we report our results on the Stanford Question Answering Dataset (SQuAD) and the General Language Understanding Evaluation (GLUE) benchmark (Wang et al., 2018). We use two versions of SQuAD: V1.1 and V2.0 (Rajpurkar et al., 2016; 2018). The GLUE is a collection of datasets/tasks for evaluating natural language understanding systems[2].

**Input/Output representations:** We follow the input/output representation setting from Devlin et al. (2019) for both pre-training and fine-tuning. We use the WordPiece (Wu et al., 2016) embeddings with a $30,000$ token vocabulary. The first token of every sentence is always a special classification token ([CLS]). The sentences are differentiated with a special token ([SEP]).

**Evaluation:** In pre-training, BERT considers two objectives: masked language modeling (MLM) and next sentence prediction (NSP). For MLM, a random sample of the tokens in the input sequence is selected and replaced with the special token ([MASK]). The MLM objective is a cross-entropy loss on predicting the masked tokens. NSP is a binary classification loss for predicting whether two segments follow each other in the original text. In pre-training, we use MLM and NSP as training objectives to pre-train, retrain the BERT model, and as metrics to evaluate the BERT model . In fine-tuning, F1 scores are reported for SQuAD, QQP and MRPC. Matthew's Corr and Pearson-Spearman Corr are reported for CoLA and SST2 respectively. Accuracy scores are reported for the other tasks.

All the experiments execute on one Google Cloud TPU V3-512 cluster, three Google Cloud TPU V2-512 clusters and 110 Google Cloud TPU V3-8/V2-8 instances.

---

[1]https://github.com/google-research/bert

[2]The datasets/tasks are: CoLA (Warstadt et al., 2018), Stanford Sentiment Treebank (SST) (Socher et al., 2013), Microsoft Research Paragraph Corpus (MRPC) (Dolan & Brockett, 2005), Semantic Texual Similarity Benchmark (STS) (Agirre & Soroa, 2007), Quora Question Pairs (QQP), Multi-Genre NLI (MNLI) (Williams et al., 2017), Question NLI (QNLI) (Rajpurkar et al., 2016), Recognizing Textual Entailment (RTE) and Winograd NLI(WNLI) (Levesque et al., 2012).

**Baseline:** As there is no public effective BERT pruning method, we use the proposed NIP pruning method on BERT as the baseline method. Th detail of NIP is shown in Appendix A. The progressive pruning ratio is $\nabla p = 10\%$ (prune 10% more weights in each iteration). Starting from the official $\text{BERT}_{\text{BASE}}$, we use 9 iterations. In each iteration $t$ of NIP, we get the sparse $\text{BERT}_{\text{BASE}}$ with specific sparsity, as $(\mathbf{w}^t; \mathcal{S}_{\mathbf{w}^t})$. Then we retrain the sparse $\text{BERT}_{\text{BASE}}$ $\mathbf{w}^t$ over the sparsity $\mathcal{S}_{\mathbf{w}^t}$. In the retraining process, the initial learning rate is $2 \cdot 10^{-5}$, the batch size is 1024 and the retraining lasts for $10,000$ steps (around 16 epochs). For the other hyperparameters, we follow the original BERT paper Devlin et al. (2019). In each iteration, the well retrained sparse $\text{BERT}_{\text{BASE}}$ is the starting point for the fine-tuning tasks and the next iteration.

## 4.2 REWEIGHED PROXIMAL PRUNING (RPP)

We apply the proposed Reweighted Proximal Pruning (RPP) method on both $\text{BERT}_{\text{BASE}}$ and $\text{BERT}_{\text{LARGE}}$, and demonstrate performance improvement. Detailed process of RPP is in Appendix B.

For $\text{BERT}_{\text{BASE}}$, we use the hyperparameters exactly the same with our experiments using NIP. The initial learning rate is $\lambda = 2 \cdot 10^{-5}$ and the batch size is 1024. We iterate the RPP for six times ($T$=6), and each iteration lasts for $100,000$ steps (around 16 epochs). The total number of epochs in RPP is smaller than NIP when achieving 90% sparsity ($96 < 144$). There is no retraining process in RPP. We set $\gamma \in \{10^{-2}, 10^{-3}, 10^{-4}, 10^{-5}\}$ and $\epsilon = 10^{-9}$ in Algorithm 1. Recall that RPP reduces to $\ell_1$ sparse training as $t=1$.

In Figure 2, we present the accuracy versus the pruning ratio for pre-training objectives MLM and NSP, and fine-tuning task SQuAD 1.1. Here we compare RPP with NIP. Along with the RPP continuing to iterate, the performance of RPP becomes notably higher than NIP for both the pre-training task and the fine-tuning task. The gap further increases as the RPP iterates more times. In Figure 2, we find that the NSP accuracy is very robust to pruning. Even when 90% of the weights are pruned, the NSP accuracy keeps above 95% in RPP algorithm and around 90% in NIP algorithm. For MLM accuracy and SQuAD F1 score, the performance drops quickly as the prune ratio increases. RPP slows down the decline trend to a great extent. On SQuAD 1.1 dataset/task, RPP keeps the F1 score of $\text{BERT}_{\text{BASE}}$ at 88.5 (0 degradation compared with original BERT) at $41.2\%$ prune ratio, while the F1 score of $\text{BERT}_{\text{BASE}}$ applied with NIP drops to 84.6 (3.9 degradation) at $40\%$ prune ratio. At $80\%$ prune ratio, RPP keeps the F1 score of $\text{BERT}_{\text{BASE}}$ at 84.7 (3.8 degradation), while the F1 score of $\text{BERT}_{\text{BASE}}$ applied with NIP drops to 68.8 (19.7 degradation compared with the original BERT). In addition to the fine-tuning task of SQuAD 1.1, the other transfer learning tasks show the same trend (RPP consistently outperforms NIP) and the detailed results are reported in Appendix C.

For $\text{BERT}_{\text{LARGE}}$, we use the hyperparameters exactly the same with our experiments using NIP except for the batch size. The initial learning rate is $2 \cdot 10^{-5}$ and the batch size is 512. We iterate the RPP for four times ($T$=6), and each iteration lasts for $100,000$ steps (around 8 epochs). There is no retraining process either. We set $\gamma \in \{10^{-2}\ 10^{-3}\ 10^{-4}\ 10^{-5}\}$ and $\epsilon = 10^{-9}$ in Algorithm 1. The experimental results about pruning $\text{BERT}_{\text{LARGE}}$ and then fine-tuning are shown in Table 1.

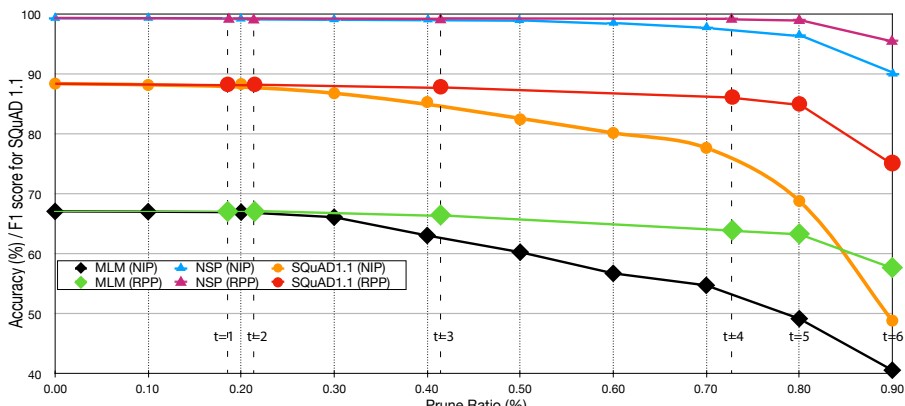

**Figure 2:** Evaluate the performance of pruned $\text{BERT}_{\text{BASE}}$ using NIP and RPP, respectively (MLM and NSP accuracy on pre-training data and F1 score of fine-tuning on SQuAD 1.1 are reported).

**Table 1:** $\text{BERT}_{\text{LARGE}}$ pruning results on a set of transfer learning tasks. The degradation is contrasted with the original BERT (without pruning) for transfer learning.

| Method | Prune Ratio(%) | SQuAD 1.1 | QQP | MNLI | MRPC | CoLA |
|---|---|---|---|---|---|---|
| NIP | 50.0 | 85.3 (-5.6) | 85.1 (-6.1) | 77.0 (-9.1) | 83.5 (-5.5) | 76.3 (-5.2) |
| | 80.0 | 75.1 (-15.8) | 81.1 (-10.1) | 73.81 (-12.29) | 68.4 (-20.5) | 69.13 (-12.37) |
| **RPP** | 59.3 | 90.23 (-0.67) | 91.2 (-0.0) | 86.1 (-0.0) | 88.1 (-1.2) | 82.8 (+1.3) |
| | 88.4 | 81.69 (-9.21) | 89.2 (-2.0) | 81.4 (-4.7) | 81.9 (-7.1) | 79.3 (-2.2) |

| Method | Prune Ratio(%) | SQuAD 2.0 | QNLI | MNLIM | SST-2 | RTE |
|---|---|---|---|---|---|---|
| NIP | 50.0 | 75.3 (-6.6) | 90.2 (-1.1) | 82.5 (-3.4) | 91.3 (-1.9) | 68.6 (-1.5) |
| | 80.0 | 70.1 (-11.8) | 80.5 (-10.8) | 78.4 (-7.5) | 88.7 (-4.5) | 62.8 (-7.3) |
| **RPP** | 59.3 | 81.3 (-0.6) | 92.3 (+1.0) | 85.7 (-0.2) | 92.4 (-0.8) | 70.1 (-0.0) |
| | 88.4 | 80.7 (-1.2) | 88.0 (-3.3) | 81.8 (-4.1) | 90.5 (-2.7) | 67.5 (-2.6) |

### 4.3 Visualizing Attention Pattern in BERT

We visualize the sparse pattern of the kernel weights in sparse BERT model applied with RPP, and present several examples in Figure 3. Since we directly visualize the value of *identical universal sparsity* $\mathcal{S}_{\mathbf{w}}$ without any auxiliary function like activation map, the attention pattern is universal and data independent.

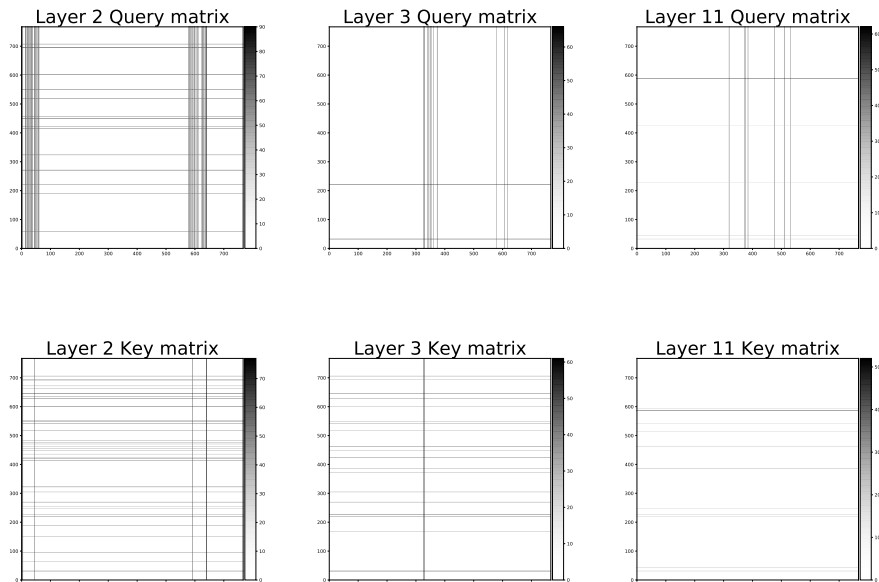

**Figure 3:** Visualization of sparse pattern $\mathcal{S}$ in pruned $\text{BERT}_{\text{BASE}}$ model $\mathbf{w}$. We sample 6 matrices (3 query matrices at the top row and 3 key matrices at the bottom row) from layer 2, layer 3 and layer 11 in the sparest pruned $\text{BERT}_{\text{BASE}}$.

BERT's model architecture is a multi-layer, bidirectional transformer encoder based on the original implementation (Vaswani et al., 2017). Following (Vaswani et al., 2017), the transformer architecture is based on "scaled dot-product attention." The input consists of queries, keys and values, denoted as matrices $Q$, $K$ and $V$, respectively. The output of attention model is computed as

$$\text{Attention}(Q, K, V) = \text{softmax}\left(\frac{QK^T}{\sqrt{d_k}}\right)V \tag{9}$$

where $d_k$ is the dimension. We visualize the sparse matrices $Q$ and $K$ of layer 2, layer 3 and layer 11 respectively in Figure 3. From Figure 3, we have the following observations and analyses.

**Structured pattern:** Figure 3 demonstrates the structured pattern of non-zero weights in a pruned transformer block. More specifically, we found that the pruned $Q$ and $K$ matrices within each transformer yield interesting group-wise structures (column-wise non-sparsity for query matrix and row-wise non-sparsity for key matrix). Interestingly, we obtained these structured sparse patterns from our proposed RPP, an irregular pruning method (namely, no group-wise sparsity is penalized). This is different from the irregular pruning on image classifiers, and thus shows the specialty of pruning on language models. We also believe that the use of reweighted $\ell_1$ approach matters to find these fine-grained sparse patterns. Note that the structured sparsity pattern is more friendly to hardware implementation and acceleration than the non-structured pattern.

**Semantic interpretation:** The structured pattern found by RPP (visualized in Figure 3) has the following semantic interpretation. What might the large-scale language representation learn? The answer becomes clear after the language representation is pruned by RPP. From the perspective of attention mechanism, the query matrix $Q$ (column-wise non-sparsity) mainly models the attention information inside each sequence, while the key matrix $K$ (row-wise non-sparsity) mainly models the attention information between different sequences in the context.

### 4.4 $t$-SNE VISUALIZATION

$t$-Distributed Stochastic Neighbor Embedding ($t$-SNE) is a technique for dimensionality reduction that is particularly well suited for the visualization of high-dimensional datasets (Maaten & Hinton, 2008). Pre-trained word embeddings are an integral part of modern NLP systems (Devlin et al., 2019) and one contribution of BERT is the pre-trained contextual embedding. Hence, we visualize word embedding in the original BERT model and the BERT model applied with RPP in Figure 4 using $t$-SNE. Since BERT is different from commonly-studied image classifier in network pruning, we would like to examine if pruning on BERT will lead to significant change on the low-dimensional manifold of the language representation. From Figure 4, we obtain the following observations and insights.

**Low-dimensional manifold:** Figure 4 illustrates that, for both original BERT and BERT pruned with RPP, the low-dimensional manifolds of the language representations are similar, showing the similar projection. Taking the specific word "intelligent" in Figure 4 as an example, the distribution of specific words and corresponding nearest words at the low-dimensional manifold (calculated using cosine/Euclidean distance) remain the high degree of similarity. This implies that the BERT applied with RPP keeps most of the language representation information similar to that from the original BERT.

**Linguistic interpretation of proper noun:** There is one salient ribbon on the upper left of the macroscopical t-SNE visualization of word embeddings in either the original BERT or the pruned BERT through RPP. Each point in the ribbon represents a year number in annals. There is also one salient short line on the lower left of the macroscopical t-SNE visualization of word embeddings in either the original BERT or the BERT applied with RPP. Each point in most of the lines represents an age number. Other proper nouns also reveal similar characteristics. Our proposed RPP remains the embedding information of these proper nouns from the perspective of linguistic interpretation.

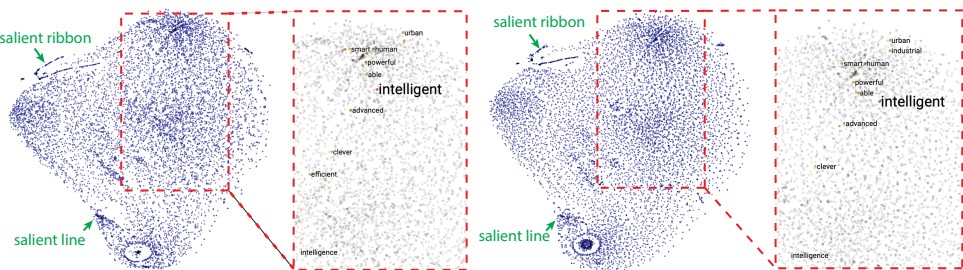

**Figure 4:** $t$-SNE visualization of word embeddings in the original BERT model and the pruned BERT model using RPP. From left to right: $t$-SNE of original BERT embedding, together with an enlarging region around word "intelligent"; $t$-SNE of embedding in pruned BERT, together with an enlarging region. These visualizations are obtained by running t-SNE for 1000 steps with perplexity=100.

## 5 Conclusions and Future Work

This paper presents the pruning algorithm *RPP*, which achieves the first effective weight pruning result on large pre-trained language representation model - BERT. RPP achieves $59.3\%$ weight sparsity without inducing the performance loss on both pre-training and fine-tuning tasks. We spotlight the relationship between the pruning ratio of the pre-trained DNN model and the performance on the downstream multi-task transfer learning objectives. We show that many downstream tasks except SQuAD allows at least $80\%$ pruning ratio compared with $59.3\%$ under task SQuAD. Our proposed Reweighted Proximal Pruning provides a new perspective to analyze what does a large language representation (BERT) learn.

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

## A  ALGORITHM OF NEW ITERATIVE PRUNING

Algorithm 2 shows the detail process of our proposed NIP algorithm.

---
**Algorithm 2** New Iterative Pruning (NIP) algorithm
---
1: Input: Initial model weights $\mathbf{w}$, initial prune ratio $p = 0\%$, progressive prune ratio $\nabla p$
2: **for** $t = 1, 2, \ldots, T$ **do**
3:     $\mathbf{w} = \mathbf{w}^{(t-1)}$
4:     Sample batch of data from the pre-training data
5:     Obtain sparsity $\mathcal{S}_{\mathbf{w}}$ through hard threshold pruning, prune ratio $p^t = t \cdot \nabla p$
6:     Retrain $\mathbf{w}$ over sparsity constraint $\mathcal{S}_{\mathbf{w}}$
7:     **for** all tasks in $\{\mathcal{T}_i\}$ **do**
8:         Fine-tune $\mathbf{z}_i(\mathbf{w}; \mathcal{S}_{\mathbf{w}})$ over sparsity $\mathcal{S}_{\mathbf{w}}$ (if the desired prune ratio $p^t$ has been reached for downstream task $i$)
9:     **end for**
10: **end for**
---

## B  ALGORITHM OF REWEIGHTED PROXIMAL PRUNING (RPP)

Algorithm 3 shows the detail process of our enhanced AdamW (Loshchilov & Hutter, 2018) with proximal operator.

---
**Algorithm 3** Our enhanced AdamW (Loshchilov & Hutter, 2018) with proximal operator
---
1: **Given** $\alpha = 0.001, \beta_1 = 0.9, \beta_2 = 0.999, \epsilon = 10^{-6}, \lambda \in \mathbb{R}$
2: **Initialize** time step $k \leftarrow 0$, parameters of pre-trained model $\mathbf{w}$, first moment vector $\mathbf{m}_{t=0} \leftarrow \mathbf{0}$, second moment vector $\mathbf{v}_{t=0} \leftarrow \mathbf{0}$, schedule multiplier $\eta_{k=0} \in \mathbb{R}$
3: **repeat**
4:     $k \leftarrow k + 1$
5:     $\nabla f_k(\mathbf{w}_{k-1}) \leftarrow \text{SelectBatch}(\mathbf{w}_{k-1})$
6:     $\boldsymbol{g}_k \leftarrow \nabla f_k(\mathbf{w}_{k-1})$
7:     $\boldsymbol{m}_k \leftarrow \beta_1 \boldsymbol{m}_{k-1} + (1 - \beta_1)\boldsymbol{g}_k$
8:     $\boldsymbol{v}_k \leftarrow \boldsymbol{\beta}_2 \boldsymbol{v}_{k-1} + (1 - \beta_2)\boldsymbol{g}_k^2$
9:     $\hat{m}_k \leftarrow \boldsymbol{m}_k / (1 - \beta_1^k)$
10:    $\hat{v}_k \leftarrow \boldsymbol{v}_k / (1 - \beta_2^k)$
11:    $\eta_k \leftarrow \text{SetScheduleMultiplier}(k)$
12:    $\mathbf{a} \leftarrow \mathbf{w}_{k-1} - \eta_k \left(\alpha \hat{m}_k / (\sqrt{\hat{v}_k} + \epsilon) + \lambda \mathbf{w}_{k-1}\right)$
13:    $\mathbf{w}_k \leftarrow \text{prox}_{\lambda_k, rw-\ell_1}(\mathbf{a})$
14: **until** stopping criterion is met
15: **return** optimized sparse model $\mathbf{w}$ in pre-training
---

# C    DOWNSTREAM TRANSFER LEARNING TASKS

As we mentioned in our main paper, we prune the pre-trained BERT model (using NIP and RPP) and then fine-tune the sparse pre-trained model to different down-stream transfer learning tasks. In this section, we exhibit the performance of pruned BERT using NIP and RPP on a wide range of downstream transfer learning tasks to demonstrate our conclusions in the main paper.

Through finetuning the pruning BERT over different downstream tasks, we found that SQuAD the most sensitive to the pruning ratio, showing an evident performance drop after $80\%$ pruning ratio. By contrast, the pruning can be made more aggressively when it is evaluated under other finetuning tasks. This is not surprising, since SQuAD is a much harder Question Answering (QA) tasks, than other simple classification tasks with limited solution space.

On the other hand, as the prune ratio of the pre-trained BERT increases, the performances on different transfer learning tasks descend generally. The descending ranges differ in different downstream transfer learning tasks. The descending range on SQuAD is the largest. Our proposed RPP mitigates the descending trend on all downstream transfer learning tasks to a great extent, compared with NIP. The intrinsic reason of this descending trend is left to future work.

## C.1    QQP

Quora Question Pairs is a binary classification task where the goal is to determine if two questions asked on Quora are semantically equivalent.

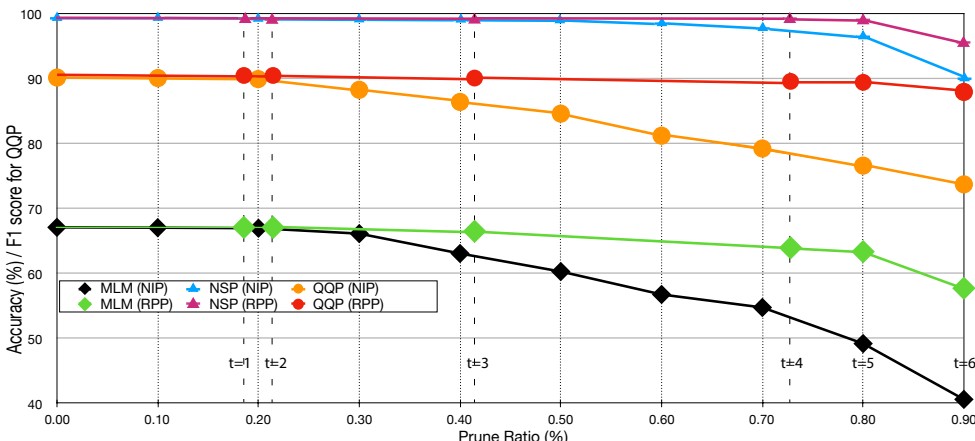

**Figure A1:** Evaluate the performance of pruned $\text{BERT}_{\text{BASE}}$ using NIP and RPP, respectively (MLM and NSP accuracy on pre-training data and F1 score of fine-tuning on QQP are reported).

**Finetuning setting:** for fine-tuning on QQP, we set learning rate $\lambda = 2 \cdot 10^{-5}$, batch size 32 and fine tuned for 3 epochs.

## C.2 MRPC

Microsoft Research Paraphrase Corpus consists of sentence pairs automatically extracted from online news sources, with human annotations for whether the sentences in the pair are semantically equivalent. Dolan & Brockett (2005)

**Finetuning setting:** for fine-tuning on MRPC, we set learning rate $\lambda = 2 \cdot 10^{-5}$, batch size 32 and fine-tune for 3 epochs.

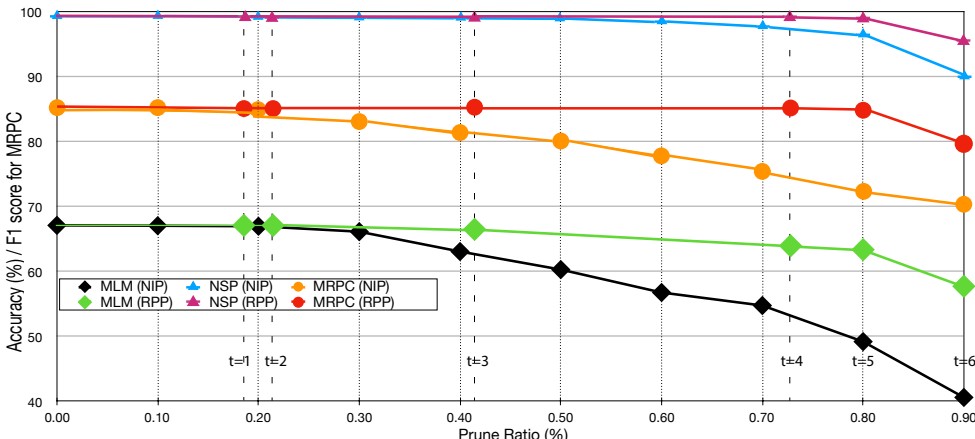

**Figure A2:** Evaluate the performance of pruned $\text{BERT}_{\text{BASE}}$ using NIP and RPP, respectively (MLM and NSP accuracy on pre-training data and F1 score of fine-tuning on MRPC are reported).

## C.3 MNLI

Multi-Genre Natural Language Inference is a large-scale, crowdsourced entailment classification task Williams et al. (2017). Given a pair of sentences, the goal is to predict whether the second sentence is an entailment, contradiction, or neutral with respect to the first one.

**Finetuning setting:** for fine-tuning on MNLI, we set learning rate $\lambda = 2 \cdot 10^{-5}$, batch size 32 and fine-tune for 3 epochs.

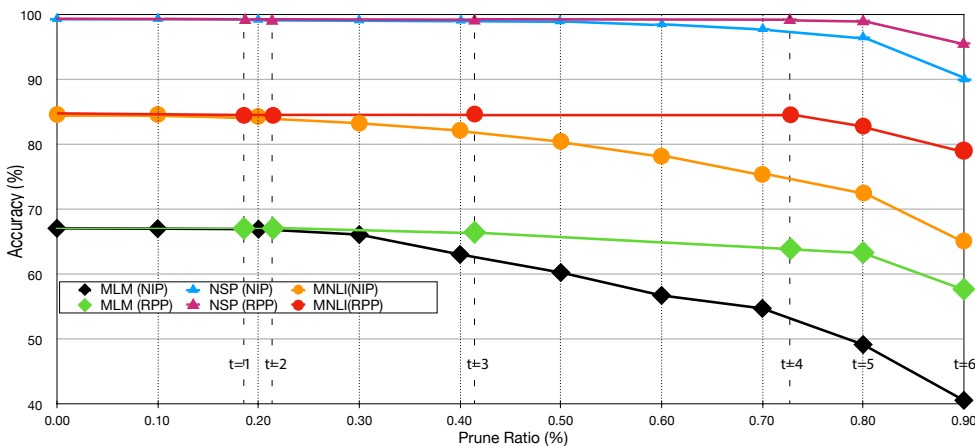

**Figure A3:** Evaluate the performance of pruned $\text{BERT}_{\text{BASE}}$ using NIP and RPP, respectively (MLM and NSP accuracy on pre-training data and accuracy of fine-tuning on MNLI are reported).

## C.4 MNLIM

Multi-Genre Natural Language Inference has a separated evaluation MNLIM. Following (Devlin et al., 2019), the fine-tuning process on MNLIM is separated from MNLI. So we present our results on MNLIM in this subsection.

**Finetuning setting:** for fine-tuning on MNLIM, we set learning rate $\lambda = 2 \cdot 10^{-5}$, batch size 32 and fine-tune for 3 epochs.

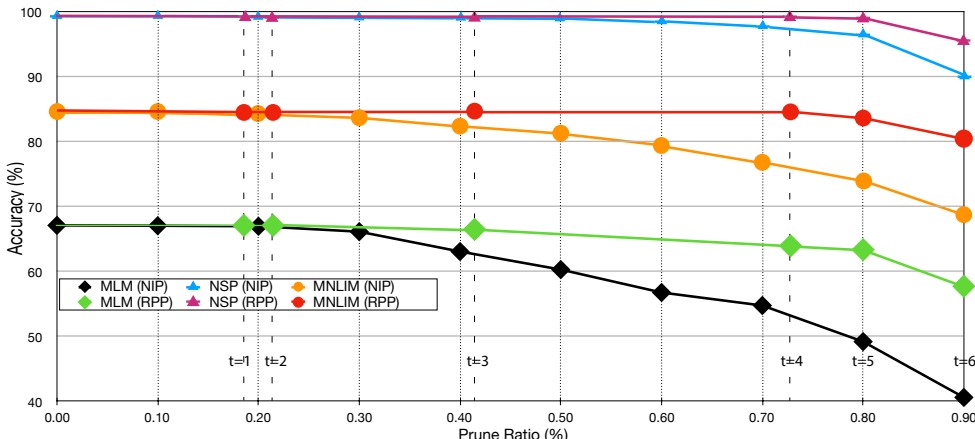

**Figure A4:** Evaluate the performance of pruned $\text{BERT}_{\text{BASE}}$ using NIP and RPP, respectively (MLM and NSP accuracy on pre-training data and accuracy of fine-tuning on MNLIM are reported).

## C.5 QNLI

Question Natural Language Inference is a version of the Stanford Question Answering Dataset (Rajpurkar et al., 2016) which has been converted to a binary classification task Wanget al., 2018a). The positive examples are (question, sentence) pairs which do contain the correct answer, and the negative examples are (question, sentence) from the same paragraph which do not contain the answer.

**Finetuning setting:** for fine-tuning on QNLI, we set learning rate $\lambda = 2 \cdot 10^{-5}$, batch size 32 and fine-tune for 3 epochs.

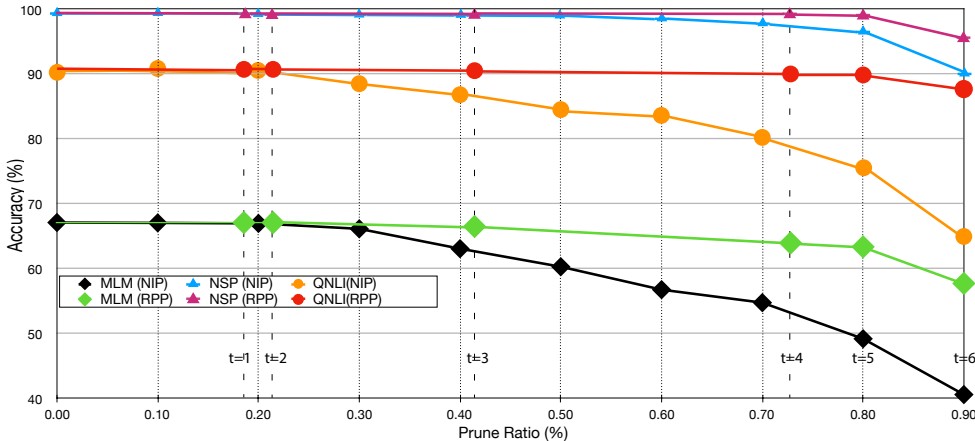

**Figure A5:** Evaluate the performance of pruned $\text{BERT}_{\text{BASE}}$ using NIP and RPP, respectively (MLM and NSP accuracy on pre-training data and accuracy of fine-tuning on QNLI are reported).

## C.6 SST-2

The Stanford Sentiment Treebank is a binary single-sentence classification task consisting of sentences extracted from movie reviews with human annotations of their sentiment(Socher et al., 2013).

**Finetuning setting:** for fine-tuning on SST-2, we set learning rate $\lambda = 2 \cdot 10^{-5}$, batch size 32 and fine-tune for 3 epochs. To consistent with the GLUE benchmark (Wang et al., 2018), the Pearson-Spearman Corr score is reported.

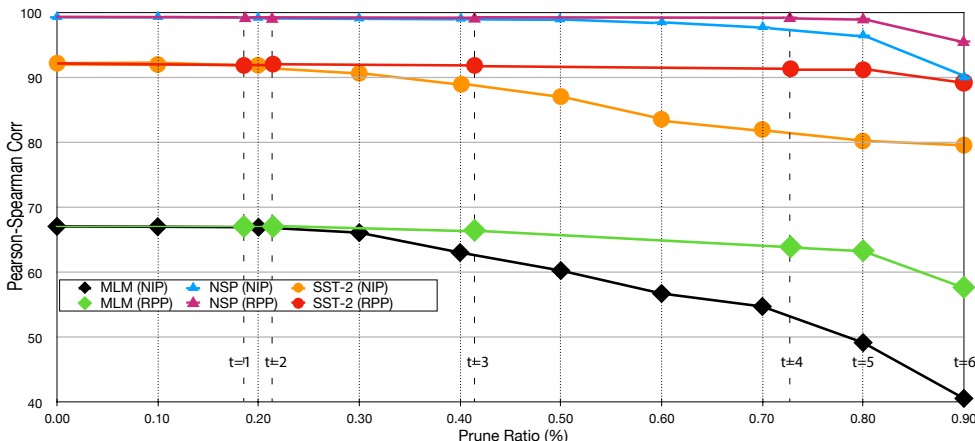

**Figure A6:** Evaluate the performance of pruned $\text{BERT}_{\text{BASE}}$ using NIP and RPP, respectively (MLM and NSP accuracy on pre-training data and accuracy of fine-tuning on SST-2 are reported).

## C.7 CoLA

The Corpus of Linguistic Acceptability is a binary single-sentence classification task, where the goal is to predict whether an English sentence is linguistically acceptable or not (Warstadt et al., 2018).

**Finetuning setting:** for fine-tuning on CoLA, we set learning rate $\lambda = 2 \cdot 10^{-5}$, batch size 32 and fine-tune for 3 epochs. To consistent with the GLUE benchmark (Wang et al., 2018), the Matthew's Corr score is reported.

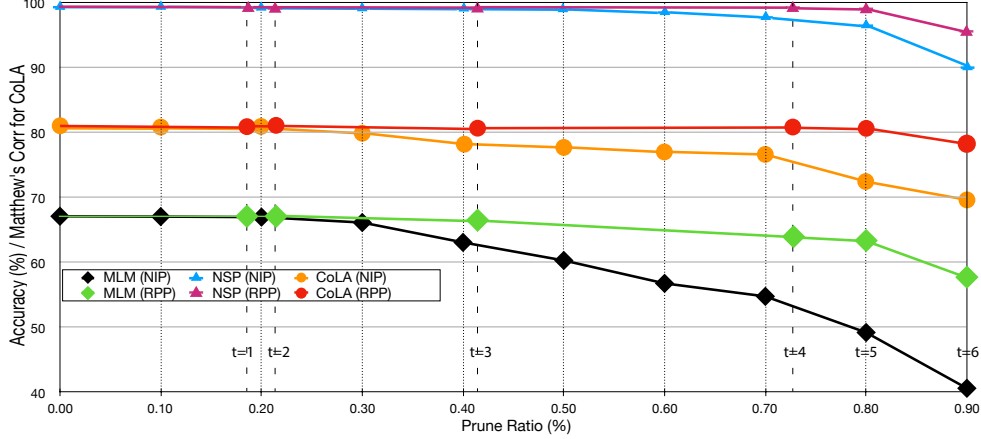

**Figure A7:** Evaluate the performance of pruned $\text{BERT}_{\text{BASE}}$ using NIP and RPP, respectively (MLM and NSP accuracy on pre-training data and accuracy of fine-tuning on CoLA are reported).

## D  NON CONVERGENCE OF PRUNING BERT USING PREVIOUS METHODS

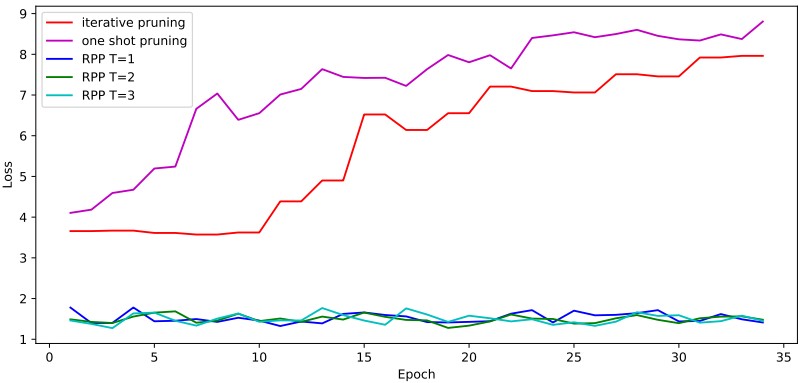

**Figure A8:** Training loss curve of applying iterative pruning and RPP on BERT

As we mentioned in our main paper, we investigate a series of pruning techniques to prune BERT, include the iterative pruning method (Han et al., 2015) and the one-shot pruning (Liu et al., 2019b). However, most of the previous pruning techniques requires to directly optimize the $\ell_1/\ell_2$ regularization using the back-propagation based gradient update in the original training of DNN models. We execute a school of experiments and find that, this kind of method to optimize regularization might not be compatible with BERT. We show the experiment results about this incompatibility in this section. For the sake of fair comparison, we not only adopt the same hyperparameters (in our experiments about NIP and RPP) on iterative pruning and one-shot pruning, we execute a wide set of hyperparamters to make the iterative pruning and one-shot pruning work. We set the learning $\lambda \in \{2 \cdot 10^{-4}, 10^{-4}, 5 \cdot 10^{-5}, 3 \cdot 10^{-5}, 2 \cdot 10^{-5}, 1 \cdot 10^{-5}, 1 \cdot 10^{-6}, 1 \cdot 10^{-7}, 1 \cdot 10^{-8}\}$, batch size $B \in \{256, 512, 1024, 2048\}$. We execute the same hyperparameters (with NIP and RPP) and attempt more hyperparameters on the iterative pruning and one-shot pruning, but iterative and one-shot pruning could not converge to a valid solution. Figure A8 illustrates training loss curve of applying iterative pruning, one-shot pruning and RPP on BERT. It is clear that iterative pruning and one-shot pruning leads to a non-convergence result, while different settings of RPP ($T = 0, T = 1, T = 2$) converge well.

From the perspective of optimization and convergence, we make the following analysis:

The previous method, such as Iterative Pruning (IP) and one-shot pruning, relies on directly optimizing the $\ell_1$ / $\ell_2$ penalized training loss to conduct DNN pruning (this is discussed by Han et al. (2015) on iterative pruning, Section 3.1). As a result, a simultaneous back-propagation (for updating model weights) is conducted over both the original training loss as well as the non-smooth sparsity regularizer. When the penalty term is back-propagated together with the loss function, this affects the convergence of the original loss function. The convergence performance is significantly degraded for extremely large DNN model like BERT. This phenomenon is also observed in the training of BERT (ADAM weight decay) that decouples the regularization term with the original loss function, instead of using an overall back-propagation.

For the super-deep DNN model (like BERT), it becomes harder to count on the one-time back-propagation flow to solve both the original training objective and the sparse regulariztion at the same time. Loshchilov & Hutter (2018) notices this limitation and improves regularization of Adam, by decoupling weight decay regularization from the gradient-based update (Loshchilov & Hutter, 2018). AdamW is widely adopted in pretraining large language representations, e.g., BERT (Devlin et al., 2019), GPT (Radford et al., 2018) and XLNet (Yang et al., 2019). The difference is that, the decoupled weight decay in AdamW is to avoid overfitting, while our purpose is to generate sparsity. Moreover, previous algorithms with directly optimizing $\ell_1$ through the back-propagation based gradient update penalty on TPU will easily lead to the gradient NaN.

Hence we proposed New Iterative Pruning (NIP) as our working baseline. We believe that NIP works since NIP reflects the naturally progressive pruning performance without any externally introduced penalty. As a fix of IP, NIP simplifies the training objective by removing the non-smooth sparsity regularizer. This simple fix improves the convergence of the training process, and makes new iterative pruning doable for BERT. We hope that other pruning methods should not perform worse than NIP, otherwise, the effect of optimizing the newly introduced sparsity-promoting regularization is negative.

To further improve the pruning performance, we need to find a better pruning method that exploits our composite objective structure (original training loss + sparsity regularization), so that the back-propagation is not affected for the original training objective of BERT. Motivated by that, proximal gradient provides an elegant solution, which splits the updating rule into a) gradient descent over the original training loss, and b) proximal operation over non-smooth sparsity regularizers. Moreover, reweighted $\ell_1$ minimization serves as a better sparsity generalization method, which self-adjusting the importance of sparsity penalization weights. Furthermore, the incorporation of reweighted $\ell_1$ will not affect the advantage of the proximal gradient algorithm. Thanks to the closed-form solution (equation 8) of proximal operation on a weighted $\ell_1$ norm, Reweighted Proximal Pruning (RPP) is a desired pruning method on BERT model. We hope RPP proves to be effective in more kinds of DNN models in the future.

