# OpenReview forum: "Reweighted Proximal Pruning for Large-Scale Language Representation"
_ICLR.cc/2020/Conference — Reject_

### Official Review · AnonReviewer3 · 2019-10-24
**Official Blind Review #3**

**Rating:** 6

**Review:**

This paper proposes a way to compress Bert by weight pruning with  L1 minimization and proximal method. This paper is one of the first works aiming at  Bert model compression.
The authors think the traditional pruning ways can not work well for Bert model, so they propose Reweighted Proximal Pruning and conduct experiments on two different datasets. According to their results, they successfully compress 88.4% of the original Bert large model and get a reasonable accuracy.

Strong points:
1. The authors propose a new method RPP for Bert model compression.
2. The authors design experiments to show their RPP can get a very good prune ratio with reasonable accuracy.

Weak points:
1. The authors should provide a detailed and rigorous explanation for the drawback of existing pruning methods.
2. In the experiments, the authors only compare RPP with self-designed method NIP instead of any existing pruning method.  The reason they said is “these methods do not converge to a viable solution''. It would be better if they are also compared and analyzed in detail.
3. In the CoLA and QNLI datasets of Bert_large experiments, RPP can get a higher accuracy even than the original Bert_large model without pruning? This is counter-intuitive.
4. About the metrics, the authors use F1 score and accuracy, the standard metrics in the GLUE benchmark for different tasks, except for CoLA. It might make sense to also keep the metrics for CoLA consistent with GLUE benchmark for better comparison.
5. It is not clear what the authors want to express in Figure 2. The generation of the figure needs more explanation, and the results need to be better interpreted.


**Experience Assessment:**

I have read many papers in this area.

**Review Assessment: Checking Correctness Of Derivations And Theory:**

I assessed the sensibility of the derivations and theory.

**Review Assessment: Checking Correctness Of Experiments:**

I carefully checked the experiments.

**Review Assessment: Thoroughness In Paper Reading:**

I read the paper at least twice and used my best judgement in assessing the paper.

---

> ### Author Response · Authors · 2019-11-15
> **Response to Review #3**
>
> Dear Review #3,
>
> Thanks so much for going through the paper carefully and providing such valuable and positive feedback. Following the reviewer's suggestions, we have updated our manuscript. The answer to each of the specific questions is below.
>
> #Question: The authors should provide a detailed and rigorous explanation for the drawback of existing pruning methods.#
>
> Response: Thanks very much for the valuable comments. We update our paper and Appendix D to compare the loss curve of the existing pruning methods with RPP. The detailed analysis is in General Response 2 (https://openreview.net/forum?id=r1gBOxSFwr&noteId=HJlbxZg2iH) and the updated Appendix D.
>
>
>
> #Question: In the experiments, the authors only compare RPP with self-designed method NIP instead of any existing pruning method. The reason they said is, "these methods do not converge to a viable solution." It would be better if they are also compared and analyzed in detail. #
>
> Response: Thanks for the valuable comments. We compare the convergence trajectories of different pruning methods in the updated Figure A8 of Appendix D. We analysis the comparison in detail,  in the updated Appendix D and General Response 2 (https://openreview.net/forum?id=r1gBOxSFwr&noteId=HJlbxZg2iH).
>
>
>
> #Question: In the CoLA and QNLI datasets of Bert_large experiments, RPP can get a higher accuracy even than the original Bert_large model without pruning? This is counter-intuitive. #
>
> Response: Thanks very much for the valuable comments. Pruning sometimes could actually improve the performance; this phenomenon also exists when pruning other deep neural networks such as CNN. One explanation is that the promotion of weight sparsity sometimes reduces overfitting.
>
>
>
> #Question: About the metrics, the authors use F1 score and accuracy, the standard metrics in the GLUE benchmark for different tasks, except for CoLA. It might make sense to also keep the metrics for CoLA consistent with GLUE benchmark for better comparison.#
>
> Response: Thanks very much for the valuable comments. The metric for CoLA is "Matthew's Corr" https://gluebenchmark.com/tasks. We are keeping the metrics consistent with GLUE in the original Figure A7 in Appendix C. The original description about metrics in Section 4.1 is not clear. We have updated the description of metrics in Section 4.1  to make it explicitly.
>
>
>
> #Question: It is not clear what the authors want to express in Figure 2. The generation of the figure needs more explanation, and the results need to be better interpreted. #
>
> Response: Thanks very much for the valuable comments. We update our original Figure 2 as the new Figure 3 for better visualization. We provided more analysis and insights from the visualization in Section 4.3 and General Response 1 (https://openreview.net/forum?id=r1gBOxSFwr&noteId=SkxYlayniS).

---

### Official Review · AnonReviewer5 · 2019-11-01
**Official Blind Review #5**

**Rating:** 6

**Review:**


Models such as BERT are pretrained language models which provide significant improvement for different tasks, however they suffer from high huge size and complexity. This paper has proposed using proximal gradient descent to find sparse weights for BERT to reduce the number of parameters and make the model smaller. They concentrate on the drawbacks of the previous sparse-based approaches and claimed that they have convergence issues (they have provided some evidence in the appendix). therefore, they propose to use reweighed sparse method and optimise it using proximal gradient descent which provides a closed form solution for sparse constraint.

ALthough proposing a minor novelty (reweighted sparse optimization ), they have provided interesting results for both pretrained structure and fine-tuning for several different tasks. they have also provided some visualisation for the weight matrices after sparsification.

Their results are notably stronger than simply adding the L1 regularizer to the optimisation method.

The paper is well written and easy to follow with nearly comprehensive related work.

However, there are some drawbacks:

1. They have claimed that “ To the best of our knowledge, we are the first to apply reweighted l1 and proximal algorithm in the DNN weight pruning domain, and achieve effective weight pruning on BERT. ”, however proximal optimization has been used for DNN in works like “Combined Group and Exclusive Sparsity for Deep Neural Networks, 2017”.
2. It should be explained clearly about all the matrices included in the sparsification steps, despite only saying “parameters of the model”.
3. More analysis is required on the results, specially the diagrams for fine-tuning over different datasets.
4. It is essential to compare the method with other related works for Bert and transformer compression, including quantisation-based, factorisation-based, pruning, knowledge distillation papers such as:
--Prato, Gabriele, Ella Charlaix, and Mehdi Rezagholizadeh. "Fully Quantized Transformer for Improved Translation." arXiv preprint arXiv:1910.10485 (2019).
--Tang, Raphael, et al. "Distilling Task-Specific Knowledge from BERT into Simple Neural Networks." arXiv preprint arXiv:1903.12136 (2019).
--Sanh, Victor, et al. "DistilBERT, a distilled version of BERT: smaller, faster, cheaper and lighter." arXiv preprint arXiv:1910.01108 (2019).
--Ziheng Wang, et al. "Structured Pruning of Large Language Models."


**Experience Assessment:**

I have read many papers in this area.

**Review Assessment: Checking Correctness Of Derivations And Theory:**

I assessed the sensibility of the derivations and theory.

**Review Assessment: Checking Correctness Of Experiments:**

I assessed the sensibility of the experiments.

**Review Assessment: Thoroughness In Paper Reading:**

I read the paper at least twice and used my best judgement in assessing the paper.

---

> ### Author Response · Authors · 2019-11-15
> **Response to Review #5 (Part A)**
>
> Dear Reviewer #5,
>
> Thanks so much for going through the paper carefully and providing such valuable and positive feedback. We make the response as below:
>
> #Question: They have claimed that "To the best of our knowledge, we are the first to apply reweighted l1 and proximal algorithm in the DNN weight pruning domain, and achieve effective weight pruning on BERT.", however proximal optimization has been used for DNN in works like "Combined Group and Exclusive Sparsity for Deep Neural Networks, 2017". #
>
> Response: Thanks for pointing out this reference. We will cite it in the related work and tone down our claim on the application of proximal gradient algorithm to network pruning. However, we would like to highlight some differences between our work on pruning BERT and (Yoon et al.) on pruning CNNs.
> First, different from CNNs, BERT adopts a pre-training + fine-tuning framework. We found the sparse pattern by pruning only on the pre-training task and then evaluated the pruning pattern on different fine-tuning tasks. We show that a highly-pruned pre-trained BERT (to the best of our knowledge, the 80% pruning ratio is significant for BERT pruning) can adapt to all down-stream tasks without losing the fine-tuning accuracy.
> Another key difference with (Yoon et al.), the use of reweighted $\ell_1$ is essential to find a sparser and a more structured pruned BERT. This conclusion is supported by the comparison between our approach and NIP, as well as the sparse patterns in the revised Figure 3.
>
>
>
> #Question: It should be explained clearly about all the matrices included in the sparsification steps, despite only saying "parameters of the model" #
>
> Response: Thanks very much for the valuable comments! In the revision, we make our description as clear as possible. It is mentioned in Section 4.1 that the weights that considered for network pruning are the parameters of transformer blocks (12 blocks in BERT base, and 24 blocks in BERT large), since these parameters take up more than 97% weights of the entire BERT. We will update our paper (Section 4.1) to make this part more explicitly.
>
>
>
> #Question: More analysis is required on the results, specially the diagrams for fine-tuning over different datasets.#
>
> Response: Thanks very much for the valuable comments. We make the following analysis and update our paper.
> In the revision, we make a more detailed analysis of the results that we obtained. Through fine-tuning the pruning BERT over different downstream tasks, we found that SQuAD the most sensitive to the pruning ratio, showing an evident performance drop after 80% pruning ratio. By contrast, the pruning can be made more aggressively when it is evaluated under other fine-tuning tasks. This is not surprising, since SQuAD is a much harder Question Answering (QA) tasks,  than other simple classification tasks with limited solution space.
> On the other hand, as the prune ratio of the pre-trained BERT increases, the performances on different transfer learning tasks descend generally. The descending ranges differ in different transfer learning tasks. The descending speed on SQuAD is the fastest. Our proposed RPP mitigates the descending trend on all downstream transfer learning tasks to a great extent, compared with NIP.
> We have an overview diagram about the pre-training and fine-tuning of BERT in Appendix A of the first submission. Thanks to your advice, we have updated the previous diagram to Section 1 of the paper, to increase the readability of our paper. We have extended more analysis about our results and extended more description about fine-tuning on each downstream task in the updated Appendix C.

---

> ### Author Response · Authors · 2019-11-15
> **Response to Review #5 (Part B)**
>
> Dear Reviewer #5,
>
> Thanks so much for going through the paper carefully and providing such valuable and positive feedback. We make the response as below:
>
> #Question: It is essential to compare the method with other related works for Bert and Transformer compression, including quantisation-based, factorisation-based, pruning, knowledge distillation papers such as:
> --Prato, Gabriele, Ella Charlaix, and Mehdi Rezagholizadeh. "Fully Quantized Transformer for Improved Translation." arXiv preprint arXiv:1910.10485 (2019).
> --Tang, Raphael, et al. "Distilling Task-Specific Knowledge from BERT into Simple Neural Networks." arXiv preprint arXiv:1903.12136 (2019).
> --Sanh, Victor, et al. "DistilBERT, a distilled version of BERT: smaller, faster, cheaper and lighter." arXiv preprint arXiv:1910.01108 (2019).
> --Ziheng Wang, et al. "Structured Pruning of Large Language Models." #
>
> Response: Thanks so much for the valuable comments and for pointing out these references.
> These are excellent works, especially for the factorization based and knowledge distillation based methods.
> Among the mentioned papers, the most relevant one is (Wang et al.), which although pruned the Transformer (basic structure block of BERT), and extend their results to Transformer-XL, by training a small dense network first. However, the pruning over BERT was not considered.
>
> (Prato et al. 2019) is about applying quantization on the Transformer model, and their method is evaluated on machine translation task. The situation about BERT is not considered in their paper. Besides, our proposed RPP is orthogonal to the quantization method. The quantized BERT model could be further pruned using RPP.
> (Tang et al. 2019) is about applying knowledge distilling on BERT, and then evaluated on the machine translation task. The situation about fine-tuning on SQuAD and GLUE benchmark is not considered.
>
> About DistillBERT (Sanh et al. 2019), we did not get a chance to compare our approach with it since our submission was actually before the arxiv date of DistillBERT (Oct. 2). However, we found the paper interesting, and will add it to the related work in the final version. In what follows, we highlight some differences between our work and DistillBERT. First, The pruning ratio of our work (at least 80%) is much higher than the model size reduction of DistillBERT (40%). Second, RPP directly works on the original BERT model architecture, so we do NOT need to design a new student network structure like DistillBERT. Finally, the RPP weight pruning method is orthogonal to Knowledge distillation, DistillBERT has a similar structure with BERT, with six layers of transformer blocks. Besides, Besides, our proposed RPP is orthogonal to knowledge distillation. In the future, we would like to examine whether or not DistillBERT could be further compressed through our proposed algorithm.
>
> About the factorization based method on BERT, ALBERT is submitted to the ICLR2020 at the same time with us. However, we found ALBERT interesting, and will add it to the related work in the final version. In what follows, we hight some differences between our work and ALBERT. ALBERT small reduces the parameters compared with BERT through weight sharing, while the total amount of computation could not be reduced through weight sharing. Another contribution from ALBERT is the factorization of the hidden matrix in Transformer, and our proposed RPP is orthogonal to the factorization of the hidden matrix. In the future, we would like to examine whether or not ALBERT could be further pruned through our proposed RPP algorithm.

---

### Official Review · AnonReviewer4 · 2019-11-01
**Official Blind Review #4**

**Rating:** 6

**Review:**

The paper proposes a new approach to prune weights that is designed keeping large scale pre-trained language representations like BERT. Such a method is desirable for deploying such models on devices with limited memory like phones etc. Experiments on Squad and Glue datasets show that a pruned version of the model maintains high accuracy for these tasks.

Pros
1. Pretty high pruning ratios (80%) can be used for many datasets (except Squad). Its an encouraging result for low-memory requirement scenarios.

Weakness:
1. Modest technical contribution. The approach description also requires elaboration. Unclear what weights participate in the pruning objective.
2. Figure 2 is difficult to understand. The paper says "The sparse attention pattern exhibits obvious structured
distribution.", but I do not know why that is desirable/useful.
3. The t-SNE visualization appears perfunctory. What should I take away from this analysis?
4. The baseline approach NIP was derived from the IP approach of Han et al. (2015). Explanation for not using IP is that it does not converge to a "viable solution". This needs more elaboration.
5. Why not compare to teacher-student distillation approaches like DistilBERT? These approaches have the same motivation of compressing model size, though different approach than what the paper adopted.

**Experience Assessment:**

I do not know much about this area.

**Review Assessment: Checking Correctness Of Derivations And Theory:**

I assessed the sensibility of the derivations and theory.

**Review Assessment: Checking Correctness Of Experiments:**

I assessed the sensibility of the experiments.

**Review Assessment: Thoroughness In Paper Reading:**

I read the paper at least twice and used my best judgement in assessing the paper.

---

> ### Author Response · Authors · 2019-11-15
> **Response to Review #4**
>
> Dear Reviewer #4,
>
> Thanks so much for going through the paper carefully and providing such valuable and positive feedback. We make the response as below:
>
> #Question: Modest technical contribution. The approach description also requires elaboration. Unclear what weights participate in the pruning objective. #
>
> Response: Our main technical contribution lies at the use of the reweighted $\ell_1$ norm as well as the proximal gradient method to reach a high pruning ratio of BERT. To the best of our knowledge, it is the first time to show that the use of a reweighted $\ell_1$ norm helps to find the accurate sparse pattern, which maintains a high pruning ratio as well as ensures the pruned model performance. We spotlight the relationship between the pruning ratio of the pre-trained DNN model and the performance on the downstream multi-task transfer learning objectives. We show that the high pruning ratio obtained from the pre-trained BERT holds on all down-stream tasks without losing the fine-tuning accuracy.
>
> In the paper, we add more description on our approach, and provide some rationale on why our approach performs well; the details are in our General Response 2 (https://openreview.net/forum?id=r1gBOxSFwr&noteId=HJlbxZg2iH) The detailed process of our proposed RPP is in the Algorithm 2 and Algorithm 3 in our paper.
>
> It was mentioned in Section 4 that the weights considered for network pruning are the parameters of transformer blocks (12 layers/blocks in BERT base, and 24 layers/blocks in BERT large) since these parameters take up more than 97% weights of the entire BERT. We update our paper to make this part more explicitly.
>
>
>
> #Question: Figure 2 is difficult to understand. The paper says "The sparse attention pattern exhibits obvious structured distribution.", but I do not know why that is desirable/useful. #
>
> Response: Thanks very much for your valuable comments. We update our original Figure 2 as the new Figure 3 for better visualization. We provided more analysis and insights from the visualization in Section 4.3 and General Response 1 (https://openreview.net/forum?id=r1gBOxSFwr&noteId=SkxYlayniS).
>
>
>
> #Question: The t-SNE visualization appears perfunctory. What should I take away from this analysis?#
>
> Response: Thanks for the valuable comments. The original Figure 3 was used to provide a visualization of how different the language representation obtained from a pruned BERT is from the representation obtained from the original BERT. We have updated the original Figure 3 as the new Figure 4 to do better visualization. And we update a detailed analysis in Section 4.4 and General Response 3 (https://openreview.net/forum?id=r1gBOxSFwr&noteId=S1xFzHl2ir)
>
>
>
> #Question: The baseline approach NIP was derived from the IP approach of Han et al. (2015). Explanation for not using IP is that it does not converge to a "viable solution". This needs more elaboration.#
>
> Response: Thanks very much for the valuable comments. We the Figure A8 to show the non-convergence of applying IP and one-shot pruning on BERT. We elaborate on the detailed analysis in the updated Appendix E and General Response 2 (https://openreview.net/forum?id=r1gBOxSFwr&noteId=HJlbxZg2iH).
>
>
>
> #Question: Why not compare to teacher-student distillation approaches like DistilBERT? These approaches have the same motivation of compressing model size, though different approach than what the paper adopted. #
>
> Response: Thanks very much for pointing out DistillBERT. We did not get a chance to compare our approach with it since our submission was actually before the arxiv date of DistillBERT (Oct. 2). However, we found the paper interesting, and will add it to the related work in the final version. In what follows, we highlight some differences between our work and DistillBERT.
> First, The pruning ratio of our work (at least 80%) is much higher than the model size reduction of DistillBERT (40%).
> Second, RPP directly works on the original BERT model architecture, so we do NOT need to design a new student network structure like DistillBERT.
> Finally, the RPP weight pruning method is orthogonal to Knowledge distillation; DistillBERT has a similar structure with BERT, with six layers of transformer blocks. In the future, we would like to examine whether or not DistillBERT could be further compressed through our proposed algorithm.

---

### Author Response · Authors · 2019-11-15
**General Response 1 to all reviewers, regarding questions about the original Figure 2**

Dear Reviewers,

Thanks so much for going through the paper carefully and providing such positive feedback. We make the responses to the questions about original Figure 2 as below:

Questions about original Figure 2 (in the revised paper, the original Figure 2 corresponds to the updated Figure 3)

Response: In the revision, we have updated the original Figure 2 (updated as Figure 3) for better visualization, and have provided more analysis and insights from the visualization. We summarize our main points below.

 The original Figure 2 and the new one (updated Figure 3) are used to demonstrate the pattern of non-zero weights in every pruned transformer block of the pruned BERT model. More specifically, we found that the pruned Query and Key matrices within each transformer yield interesting group-wise structures (column-wise non-sparsity for Query matrix and row-wise non-sparsity for Key matrix). Interestingly, we obtained these structured sparse patterns from our irregular pruning method (namely, no group-wise sparsity is penalized). This is different from the irregular pruning on image classifiers, and thus shows the specialty of pruning on language models. We also believe that the usage of the reweighted $\ell_1$ approach matters to find these fine-grained sparse patterns.

For better visualization, in the revised paper, we present the ratio of # non-zero weights at each row/column of a key or value matrix. Based on our updated Figure 3, we make the following analysis:

Structured pattern: we observe that most of the non-zero weights in the Key matrix obey a column-wise non-sparsity pattern, while there exists a column-wise non-sparsity pattern for a value matrix to some extent. The observation is consistent among multiple transformer blocks. Note that the structured sparsity pattern is more friendly to hardware implementation and acceleration than the non-structured pattern.

Semantic interpretation: the structured pattern found by RPP (visualized in updated Figure 3) has the following semantic interpretation. What might the large-scale language representation learn? The answer becomes clear after the language representation is pruned by the desired pruning algorithm, RPP. In the perspective of attention mechanism, the Query matrix $Q$ (column-wise non-sparsity) mainly models the attention information inside each sequence, while the Key matrix $K$ (row-wise non-sparsity) mainly models the attention information between different sequences in the context.

---

### Author Response · Authors · 2019-11-15
**General Response 2 to all reviewers, regarding questions on the issue of previous pruning method**

Dear Reviewers,

Thanks so much for the valuable comments. We make the responses to the questions on the issue of previous pruning method as below:

In the revision, we have updated the figure in Appendix D for better visualization, and have provided more details about the issue of previous methods. We summarize our main points below.

a) The previous method, such as Iterative Pruning (IP) and one-shot pruning, relies on directly optimizing the $\ell_1$ / $\ell_2$ penalized training loss to conduct DNN pruning (this is discussed in the NeurIPS 2015 paper by Han et al on iterative pruning, Section 3.1). As a result, a simultaneous backpropagation (for updating model weights) is conducted over both the original training loss as well as the non-smooth sparsity regularizer. When the penalty term is backpropagated together with the loss function, this affects the convergence direction of the original loss function. The convergence performance is significantly degraded for extremely large DNN model like BERT. This phenomenon is also observed in the training of BERT (Adam Weight Decay) that decouples the regularization term with the original loss function, instead of using an overall backpropagation.

Our updated figures in Appendix D helps to illustrate this issue. IP and one-short pruning easily leads to non-convergence (we use the same hyperparameters as our NIP). Moreover, we observe that previous algorithms with directly optimizing the $\ell_1$ penalty on TPU will easily lead to the gradient NaN. Our NIP method converges much better and serves as the new baseline method.

b) We proposed New Iterative Pruning (NIP) as our worked baseline. As a fix of IP, NIP simplifies the training objective by removing the non-smooth sparsity regularizer. This simple fix improves the convergence of the training process, and make new iterative pruning doable for BERT.

c) To further improve the pruning performance, we need to find a better pruning method that exploits our composite objective structure (original training loss + sparsity regularization), so that the backpropagation is not affected for the original training objective of BERT. Motivated by that, the proximal gradient provides an elegant solution, which splits the updating rule into a) gradient descent over the original training loss, and b) proximal operation over non-smooth sparsity regularizers. Moreover, reweighted $\ell_{1}$ minimization serves as a better sparsity generalization method, which self-adjusting the importance of sparsity penalization weights. Furthermore, the incorporation of reweighted $\ell_1$ will not affect the advantage of the proximal gradient algorithm. Thanks to the closed-form solution (equation 8) of proximal operation on a weighted $\ell_1$ norm, Reweighted Proximal Pruning (RPP) is a desired pruning method on BERT model. We hope RPP proves to be effective in more kinds of DNN models in the future.

---

### Author Response · Authors · 2019-11-15
**General Response 3 to all reviewers, regarding questions about the original Figure 3**

Dear Reviewers,

Thanks so much for the valuable comments. We make the responses to the questions about the original Figure 3 as below:

Questions about the original Figure 3 (in the revised paper, the original Figure 3 corresponds to the updated Figure 4)

Response: The original Figure 3 was used to provide a visualization of how different the language representation obtained from a pruned BERT is from the representation obtained from the original BERT. Since BERT is different from commonly-studied image classifiers in network pruning, we would like to examine if pruning on BERT will lead to a significant change in the low-dimensional manifold of the language representation.

Originally the reduced dimension using t-SNE is $3$ ($x-y-z$ space), and thus we presented our results projected to $x-y$ space and $y-z$ space only. Based on the reviewer’s comments, we realize that the presentation could be misleading. In the revised paper, we consider a 2-D t-SNE to make our visualization more easily.

From the updated Figure 4, we make the following observation and analyses:

Low-dimensional manifold: for both original BERT and BERT pruned with RPP, the low-dimensional manifolds of the language representation are similar, showing a similar projection.
Taking the specific word ``intelligent" in Figure 4 as an example, the distribution of specific words and corresponding nearest words at the low-dimensional manifold (calculated using cosine/Euclidean distance) remains a high degree of similarity. This observation implies that the BERT applied with RPP keeps most of the language representation information similar to that from the original BERT.

Linguistic interpretation of proper noun: There is one salient ribbon on the upper left of the macroscopical t-SNE visualization of word embeddings in either the original BERT or the pruned BERT through RPP. Each point in the ribbon represents a year number in annals. There is also one salient short line on the lower left of the macroscopical t-SNE visualization of word embeddings in either the original BERT or the BERT applied with RPP. Each point in most of the lines represents an age number. Other proper nouns also reveal similar characteristics. Our proposed RPP remains the embedding information of these proper nouns from the perspective of linguistic interpretation.

---

### Decision · Program_Chairs · 2019-12-19

**Decision:**

Reject

**Comment:**

This paper proposes a novel pruning method for use with transformer text encoding models like BERT, and show that it can dramatically reduce the number of non-zero weights in a trained model while only slightly harming performance.

This is one of the hardest cases in my pile. The topic is obviously timely and worthwhile. None of the reviewers was able to give a high-confidence assessment, but the reviews were all ultimately leaning positive. However, the reviewers didn't reach a clear consensus on the main strengths of the paper, even after some private discussion, and they raised many concerns. These concerns, taken together, make me doubt that the current paper represents a substantial, sound contribution to the model compression literature in NLP.

I'm voting to reject, on the basis of:

- Recurring concerns about missing strong baselines, which make it less clear that the new method is an ideal choice.
- Relatively weak motivations for the proposed method (pruning a pre-trained model before fine-tuning) in the proposed application domain (mobile devices).
- Recurring concerns about thin analysis.

---

> ### Author Response · Authors · 2020-01-26
> **Reply to the decision**
>
> Although this decision is not a happy ending, we appreciate OpenReview and the transparency of ICLR, so that the community could see our work.